## [Decision Letter · Decision Letter 0]

15 Oct 2025

Dear Dr. McKinnon,

Thank you for submitting your manuscript to PLOS ONE. After careful consideration, we feel that it has merit but does not fully meet PLOS ONE’s publication criteria as it currently stands. Therefore, we invite you to submit a revised version of the manuscript that addresses the points raised during the review process.

**Dear Authors,**
**I have received comments on your research article, you would see that they are advising some changes, agreeting with the reviewer, I do suggest you to go through the suggested changes and modify your manuscript accordingly.**
**Regards**

We look forward to receiving your revised manuscript.

Kind regards,

Allah Bakhsh

Academic Editor

PLOS ONE

Journal Requirements:

The authors have no competing interests to declare.

Reviewers' comments:

Reviewer's Responses to Questions

**Comments to the Author**

1. Is the manuscript technically sound, and do the data support the conclusions?

Reviewer #1: Yes

2. Has the statistical analysis been performed appropriately and rigorously?

Reviewer #1: Yes

3. Have the authors made all data underlying the findings in their manuscript fully available?

Reviewer #1: Yes

4. Is the manuscript presented in an intelligible fashion and written in standard English?

Reviewer #1: Yes

Reviewer #1: Dear Authors,

congratulations for the study, is interesting and well write (scientific base, results and discusion). The results presentations could be improve (tables and figures). Se other suggestion inside document attached.

**Do you want your identity to be public for this peer review?** For information about this choice, including consent withdrawal, please see our Privacy Policy

Reviewer #1: **Yes:** Luis O. Viteri

---

## [Author Response · Author response to Decision Letter 1]

3 Nov 2025

On behalf of all manuscript co-authors, we feel the reviewer-requested revisions were reasonable and adopted them in the revised manuscript accordingly. We are confident that the revised manuscript will meet all criteria necessary for publication in your journal. Thank you.

---

## [Decision Letter · Decision Letter 1]

2 Dec 2025

Dear Dr. McKinnon,

Thank you for submitting your manuscript to PLOS ONE. After careful consideration, we feel that it has merit but does not fully meet PLOS ONE’s publication criteria as it currently stands. Therefore, we invite you to submit a revised version of the manuscript that addresses the points raised during the review process.

We look forward to receiving your revised manuscript.

Kind regards,

Allah Bakhsh

Academic Editor

PLOS ONE

Journal Requirements:

Additional Editor Comments:

Please see the attached file to address the minor concerns of reviewer

Reviewers' comments:

Reviewer's Responses to Questions

**Comments to the Author**

Reviewer #2: All comments have been addressed

2. Is the manuscript technically sound, and do the data support the conclusions?

Reviewer #2: Yes

3. Has the statistical analysis been performed appropriately and rigorously?

Reviewer #2: Yes

4. Have the authors made all data underlying the findings in their manuscript fully available?

Reviewer #2: Yes

5. Is the manuscript presented in an intelligible fashion and written in standard English?

Reviewer #2: Yes

Reviewer #2: The study evaluates the safety of the Vip3Cb1 insecticidal protein by relying primarily on bioinformatics, in vitro digestive stability, and a short-term acute oral toxicity study in mice using purified Vip3Cb1 protein. While the authors follow common regulatory approaches, several critical issues remain unaddressed.

My primary concerns relate to the experimental design of the in vivo study, the use of purified protein instead of whole cottonseed, the short duration of the study, and the lack of detailed toxicological endpoints, including histopathology and gene expression analyses.

These weaknesses significantly limit the strength of the authors’ safety conclusions.

I recommend minor revision, with justification added for each critical omission.

Detailed comments are attached as a separate file.

**Do you want your identity to be public for this peer review?** For information about this choice, including consent withdrawal, please see our Privacy Policy

Reviewer #2: **Yes:** Saira Azam

---

## [Author Response · Author response to Decision Letter 2]

12 Jan 2026

We welcome the critical review of our manuscript by the reviewer and have attached a rebuttal letter which responds to each point raised by the reviewer.

---

## [Decision Letter · Decision Letter 2]

23 Feb 2026

Food and feed safety of a novel Vip3C insecticidal protein derived from Paenibacillus spp. for protection of crop plants against key lepidopteran pests

PONE-D-25-37773R2

Dear Dr. McKinnon,

We’re pleased to inform you that your manuscript has been judged scientifically suitable for publication and will be formally accepted for publication once it meets all outstanding technical requirements.

Kind regards,

Allah Bakhsh

Academic Editor

PLOS One

Additional Editor Comments (optional):

Reviewers' comments:

Reviewer's Responses to Questions

**Comments to the Author**

Reviewer #2: All comments have been addressed

2. Is the manuscript technically sound, and do the data support the conclusions?

Reviewer #2: Yes

3. Has the statistical analysis been performed appropriately and rigorously?

Reviewer #2: Yes

4. Have the authors made all data underlying the findings in their manuscript fully available?

Reviewer #2: Yes

5. Is the manuscript presented in an intelligible fashion and written in standard English?

Reviewer #2: Yes

Reviewer #2: this manuscript is accepted. I also attached the response sheet. proceed this article for further processing

**Do you want your identity to be public for this peer review?** For information about this choice, including consent withdrawal, please see our Privacy Policy

Reviewer #2: **Yes:** Dr Saira Azam

---

## [Editor Report · Acceptance letter]

PONE-D-25-37773R2

PLOS One

Dear Dr. McKinnon,

I'm pleased to inform you that your manuscript has been deemed suitable for publication in PLOS One. Congratulations! Your manuscript is now being handed over to our production team.

Kind regards,

on behalf of

Dr. Allah Bakhsh

Academic Editor

PLOS One